# Efficacy and safety of rifaximin in preventing hepatic encephalopathy: A systematic review and meta-analysis

Yangyang Hu, Xing Zhang, Ying Xiao, Zhinian Wu, Yadong Wang◯*

Department of Infectious Diseases, Hebei Medical University Third Hospital, Shijiazhuang, China

* wangyadong@hebmu.edu.cn

## Abstract

Rifaximin (RFX) is recommended for the treatment of hepatic encephalopathy (HE). However, evidence on whether RFX application could yield additional benefits for preventing HE in patients with cirrhosis is limited. In this study, we aimed to assess the safety and efficacy of RFX in preventing HE. We conducted a systematic search of randomized controlled trials to evaluate the use of RFX by analyzing HE incidence, hospitalization, all-cause mortality, and adverse events. Compared with the control group, RFX had a beneficial effect on the primary prevention of HE (RR = 0.58, 95% CI: 0.50–0.68), with noncomparable effects to NADs (including lactulose and lactitol, RR = 0.65, 95% CI: 0.38–1.11), but more effective than placebo (RR = 0.57, 95% CI: 0.47–0.69). After more than 1 month of RFX treatment, the risk of HE decreased significantly (RR = 0.55, 95% CI: 0.47–0.65). In secondary prevention of HE, RFX decreased the recurrence risk (RR = 0.49, 95% CI: 0.40–0.61). RFX helped to reduce the incidence of HE after transjugular intrahepatic portosystemic stent shunt (TIPSS) (RR = 0.70, 95% CI: 0.51–0.96). In terms of adverse effects, RFX was associated with a lower risk of diarrhea than NADs (RR = 0.04, 95% CI: 0.00–0.25). So, RFX therapy is effective and well-tolerated in preventing HE, and can be used as the first choice in the prophylaxis of HE after TIPSS.

## Introduction

Hepatic encephalopathy (HE) is a neuropsychiatric disorder that results from severe hepatic dysfunction or abnormalities of a portosystemic shunt. The prevalence of overt hepatic encephalopathy (OHE) in cirrhotic patients is 10–14%, with 16–21% in decompensated cirrhosis [1], especially after transjugular intrahepatic portosystemic stent shunt (TIPSS), with an incidence of HE as high as 55% [2]. HE may lead to mild cognitive impairment in 60–80% of patients with cirrhosis, affecting their ability and quality of daily life. When cirrhotic patients develop OHE, the 1-year mortality is 64% [3] and the 5-year mortality is as high as 85% [4]. Furthermore, HE occurs in 60% of

**Data availability statement:** All relevant data are within the manuscript and its Supporting Information files.

**Funding:** This study was supported by grants from the Natural Science Foundation of Hebei Province of China (H2023206042).

**Competing interests:** The authors have declared that no competing interests exist.

**Abbreviations:** CDI, clostridium difficile infection; HE, hepatic encephalopathy; LOLA, L-ornithine-L-aspartate; NADs, nonabsorbable disaccharides; OHE, overt hepatic encephalopathy; RCTs, randomized controlled trials; RFX, rifaximin; RR, relative risk; SMD, standardized mean difference; TIPSS, transjugular intrahepatic portosystemic stent shunt; 95% CI, 95% confidence interval.

patients with acute-on-chronic liver failure and worsens the prognosis [5]. Therefore, the prevention of HE is crucial for patients with cirrhosis.

Although the pathogenesis of HE in patients with cirrhosis is incompletely elucidated, there is a general agreement that gut-derived ammonia and blood ammonia due to portosystemic shunt are closely related to the onset and recurrence of HE [6]. Rifaximin (RFX), an oral antimicrobial agent derived from rifamycin, has a broad antibacterial spectrum and can reduce ammonia production by inhibiting intestinal bacteria [7] Compared with other oral antibiotics that inhibit intestinal bacteria, RFX has the advantages of minimal absorption in the intestine (<0.4%) and low systemic adverse effects. The efficacy and safety of RFX for the prevention of HE have been assessed in a number of randomized controlled trials (RCTs), most of which tend to support the application of RFX in HE prevention. Thus, RFX has been recommended by the European Association for the Study of the Liver (EASL) and the French Association for the Study of the Liver (AFEF) as an add-on therapy for lactulose to prevent the recurrence of HE [8,9].

However, it has also been shown that compared to nonabsorbable disaccharides (NADs), such as lactulose and lactitol, RFX significantly reduces the risk of relapse within 6 months in cirrhotic patients complicated with recurrent HE [10]. Despite published Meta-analysis evaluating the effectiveness of RFX in the prevention of recurrent episodes of HE, there are only a few assessments of RFX in the prevention of HE [11]. Therefore, this study conducted Meta-analysis by searching all randomized controlled trials (RCTs) on RFX for HE prevention until December 31, 2023 to comprehensively evaluate the effectiveness and safety of RFX versus non-RFX (including NADs, other antibacterial drugs, L-ornithine-L-aspartate (LOLA)) and placebo for prevention of HE, and different doses of RFX compared to prevent HE in patients with cirrhosis, aiming to provide more accurate evidence on optimizing the clinical application of RFX for the prevention of HE.

## Materials and methods

The systematic review was conducted following the PRISMA statement (S1 File. **PRISMA** check list) [12] and in accordance with a pre-specified protocol, published on PROSPERO (registration number: CRD42023418047). Ethics approval and consent to participate were not applicable.

### Search strategy

The search was made of Pubmed, Cochrane Library, Embase, CNKI, SinoMed, VIP database, and Wanfang Medical Network. The last search update was undertaken on December 31, 2023. We used the keywords for search strategies as follows: Rifaximin, Rifamycin, Xifaxan, Hepatic Encephalopathy, Hepatic Stupor, Hepatic Coma, Portal systemic encephalopathy, and their relative Medical Subject Heading (MeSH) terms.

### Study selection and data extraction

The studies were included if they met the following criteria: 1) Participant: patients with cirrhosis, adults with an age of at least 18 years; 2) Intervention: rifaximin compared with other interventions, such as NADs, other antibiotics, LOLA, and placebo;

3) Study design: RCTs only; 4) Outcomes: including HE incidence, all-cause mortality adverse events, blood ammonia levels, hospitalization rate, etc. The exclusion criteria were as follows: Literature that does not provide clinical data or provides incomplete data, and literature providing data that cannot be converted in the analysis.

Two reviewers (Y.Y.H and X.Z) independently scrutinized searches and screened the titles, abstracts, and full-text articles to list all potentially eligible RCTs. A third reviewer (Y.X). joined to decide the final selection of RCTs if a disagreement occurred between the first two reviewers. Two review authors (Y.Y.H and X.Z) independently extracted the data from selected RCTs. Contrary opinions were resolved through discussion with the assistance of a third author (Y.D.W). The extracted data included the following: first author, publication year, study design, type of patients, study groups, sample size, treatment duration, outcomes.

### Assessment of risk of bias

The risk of bias was assessed based on the Cochrane Risk of Bias Assessment Tool in RCTs. This tool comprises the following domains: random sequence generation, allocation concealment, blinding of participants and personnel, blinding of outcome assessment, incomplete outcome data, selective reporting, and other bias. Each domain was judged as high, low, or unclear risk of bias.

### Outcome measures

The primary outcome measures were as follows: HE incidence including the proportion of patients with no previous history of HE who experienced HE during RFX treatment, the proportion of patients with a previous history of HE who developed a breakthrough episode of HE or OHE during RFX application (breakthrough episode of HE was defined as an increase from a baseline Conn score of 0 or 1 to ≥ 2 or an increase from a baseline Conn score of 0 to a Conn score of 1 plus a 1-unit increase in the asterixis grade [13]). All-cause mortality viz., the proportion of patients who died from all-cause causes during RFX treatment. The incidence of adverse events was calculated by the frequency of adverse events during RFX treatment, including dizziness or headache, abdominal pain, diarrhea, nausea or vomiting, bloating or abdominal discomfort, constipation, rash, etc.

The secondary outcome measures were as follows: the effect of RFX on blood ammonia was analyzed by blood ammonia difference before and after treatment; Hospitalization rate viz., the proportion of patients hospitalized due to underlying disease or HE episodes.

### Data synthesis and analysis

Statistical analyses were carried out using Stata (version 15). Heterogeneity was assessed by $I^2$ statistics, and significant heterogeneity was defined as $I^2 > 50.0\%$. Homogeneous data was analyzed by the fixed-effects model, while the random-effects model was used to analyze heterogeneous results. Dichotomous variables are reported as the risk ratio (RR) with 95% confidence interval (CI), and the standard mean difference with 95% CI was used for continuous variables. Statistical significance was set at $P < 0.05$. Funnel plots and Egger's test was used to evaluate publication bias. If publication bias existed, the stability of the results was investigated via the trim and filling method.

## Results

### Study selection

A total of 2875 relevant references were obtained by searching the database, of which 969 papers were duplicates. After excluding the duplicates, 1906 references were further removed based on reading the titles and abstracts. 255of the remaining 267 references were removed again for study subjects and outcome index compliance, duplicate data assessment (S2 File. < Table includes all literature and exclusion grounds>). Finally, 12 RCTs were included in this study after checking the original text [10,13–23] (Fig 1).

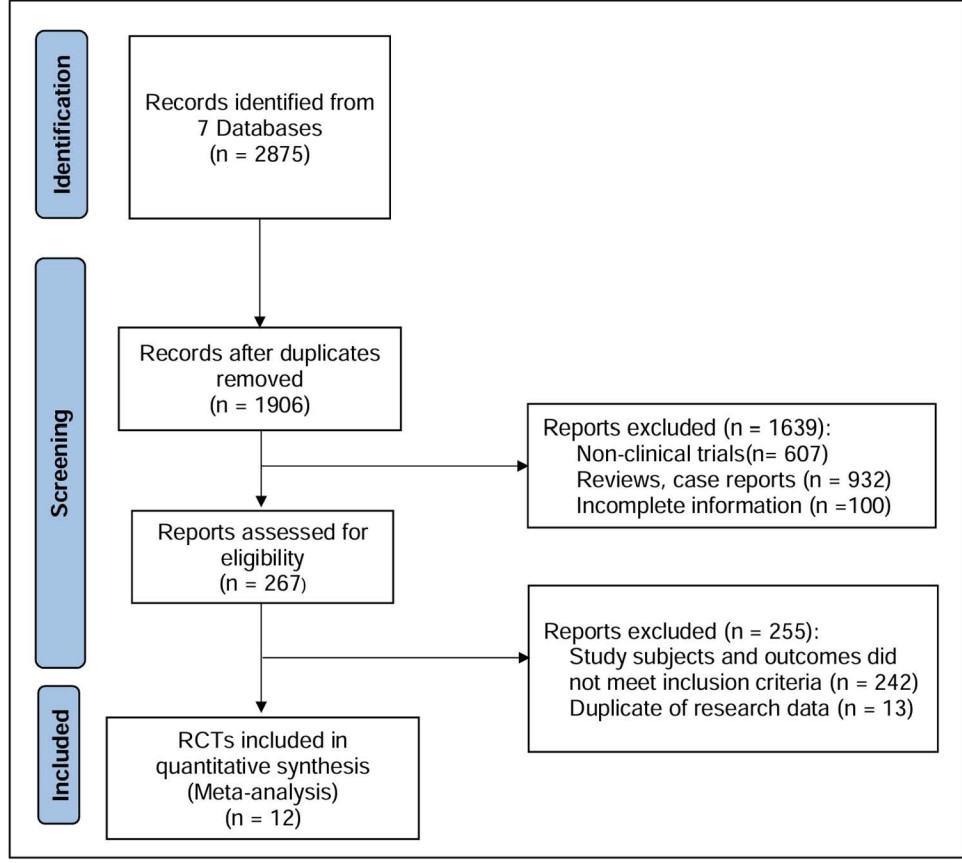

**Fig 1. Flowchart of Study Screen and Exclusion Criteria in this Meta-analysis.** 12 RCT studies met the criteria were included in this Meta analysis.

## Study Characteristics

A total of 1939 patients with cirrhosis were involved in 12 RCTs. 10 RCTs reported the effect of RFX compared with non-RFX and placebo for prevention of HE episodes (4 RCTs designed non-RFX as a control group, 4 RCTs designed placebo as a control group, 2 RCTs designed non-RFX and placebo as control groups). 720 patients were treated with RFX, 318 were in the non-RFX group, and 515 received a placebo. Two studies reported the effect of different doses of RFX on the prevention of HE, with 168 patients treated with RFX ≤ 550 mg/d and 218 patients treated with RFX 1100mg/d, respectively. 9 RCTs reported the effect of RFX application on all-cause mortality, 6 RCTs reported adverse effects in patients with cirrhosis during RFX application. 3 studies reported the effect of RFX application on blood ammonia, and 2 studies reported the hospitalization rate of patients with cirrhosis (Supplementary S1 Table ).

## Evaluation of risk bias

The Cochrane Risk of Bias Assessment Tool was used to determine the risk of bias. All of the 12 RCTs described the random sequence generation, while 6 RCTs described the allocation concealment and 5 RCTs described the blinding of participants and personnel as low risk. Most studies assessed the blinding in outcome assessment as unclear risk of bias. All of the 12 RCTs manifested a low risk of attrition, reporting or other biases (S1 Fig).

### Results of the meta-analysis

**Primary outcomes.  HE incidence:** 10 studies reported the effectiveness of RFX compared with the non-RFX and placebo in preventing HE, including 1553 patients with cirrhosis, of which 720 patients were in the RFX group, and 833 patients in the control group (including non-RFX and placebo). Meta-analysis demonstrated that compared with the control group, RFX treatment significantly reduced the incidence of HE in patients with cirrhosis (RR = 0.58, 95% CI: 0.50–0.68, $I^2$ = 31.9%, $P$ = 0.000) (Fig 2A).

Subgroup analysis based on the control drug categories indicated that RFX has noncomparable effects to NADs in reducing the risk of HE occurrence (RR = 0.65, 95% CI: 0.38–1.11, $I^2$ = 51.8%, $P$ = 0.114), but more effective than placebo (RR = 0.57, 95% CI: 0.47–0.69, $I^2$ = 0.0%, $P$ = 0.000) (Fig 2B and C). In addition, the difference between RFX and other antimicrobials (nitazoxanide and norfloxacin) in preventing HE was not statistically significant (RR = 0.91, 95% CI: 0.38–2.20, $I^2$ = 61.5%, $P$ = 0.838] (Fig 2D).

Subgroup analysis of 10 RCTs based on treatment duration showed that RFX treatment was more effective in reducing the risk of HE compared with the control group in the treatment duration >1 month subgroup (RR = 0.55 95% CI: 0.47–0.65, $I^2$ = 39.2%, $P$ = 0.000) (Fig 3A). While the treatment duration ≤1 month, the difference in HE incidence between the RFX and the control group was not statistically significant (RR = 0.84, 95% CI: 0.54–1.32, $I^2$ = 0.0%, $P$ = 0.446) (Fig 3A).

Two RCTs including 386 patients compared the effectiveness of the RFX low-dose group (≤550 mg/d) with the conventional dose group (1100 mg/d) for HE prophylaxis, with 168 patients in the low-dose group, and 218 patients in the conventional dose group, and the duration of RFX treatment was 6 months in both two RCTs. Meta-analysis showed that the difference in HE incidence between the two groups was not statistically significant (RR = 0.80, 95% CI: 0.58–1.11, $I^2$ = 49.5%, $P$ = 0.185) (Fig 3B).

2 of the 10 RCTs reported the application of RFX for primary prevention of HE in patients with cirrhosis, with 49 patients in the RFX group and 95 patients in the control groups. Meta-analysis demonstrated that RFX was comparable to the controls in primary prevention of HE (RR = 0.80, 95% CI: 0.46–1.39, $I^2$ = 0.0%, $P$ = 0.429) (Fig 4A). 5 of the 10 RCTs reported on the effectiveness of RFX for secondary prevention in patients with a history of previous HE episodes, with 396 patients in the RFX group and 437 patients in the control group. Meta-analysis demonstrated that RFX treatment has a greater advantage over the controls in reducing the risk of HE recurrence. (RR = 0.49, 95% CI: 0.40–0.61, $I^2$ = 41.0%, $P$ = 0.000) (Fig 4A).

2 of the 10 RCTs reported on the effectiveness of RFX in preventing HE after TIPSS, enrolling 236 patients, 118 patients in the RFX group, and 118 patients in the placebo group. Treatment with RFX had a beneficial effect on reducing the risk of HE after TIPSS compared with placebo (RR = 0.70, 95% CI: 0.51–0.96, $I^2$ = 0.0%, $P$ = 0.027) (Fig 4B). 2 of the 10 RCTs reported the effect of RFX on the prevention of HE in patients with cirrhosis combined with variceal bleeding, 81 patients in the RFX group, 118 patients in the control group. RFX was comparable to the controls in preventing HE in cirrhotic patients with variceal bleeding (RR = 0.79, 95% CI: 0.44–1.41, $I^2$ = 0.0%, $P$ = 0.420) (Fig 4B).

**All-cause mortality:** A total of 1493 participants with cirrhosis were included in 9 RCTs that reported on mortality during RFX administration, with 690patients in the RFX group and 803 patients in the control group. The results of both overall and subgroup analyses showed no statistical difference in mortality risk between the two groups (overall: RR = 0.98, 95% CI: 0.71–1.34, $I^2$ = 0.0%, $P$ = 0.901; RFX *vs.* non-RFX: RR = 1.02, 95% CI: 0.67–1.56, $I^2$ = 0.0%, $P$ = 0.911; RFX *vs.* placebo: RR = 0.93, 95% CI: 0.58–1.50, $I^2$ = 0.0%, $P$ = 0.776) (S2 Fig).

**Adverse events:** 6 RCTs reported the incidence of adverse events during RFX treatment such as dizziness or headache, abdominal pain, diarrhea, nausea or vomiting, bloating or abdominal discomfort, constipation, and rash. The difference in the incidence of adverse events was not statistically significant (Supplementary S2 Table ). In terms of the incidence of diarrhea, the subgroup analysis demonstrated that RFX treatment had a significantly lower incidence of diarrhea compared with NADs treatment (RR = 0.04, 95% CI: 0.00–0.25, $I^2$ = 0.0%, $P$ = 0.001).

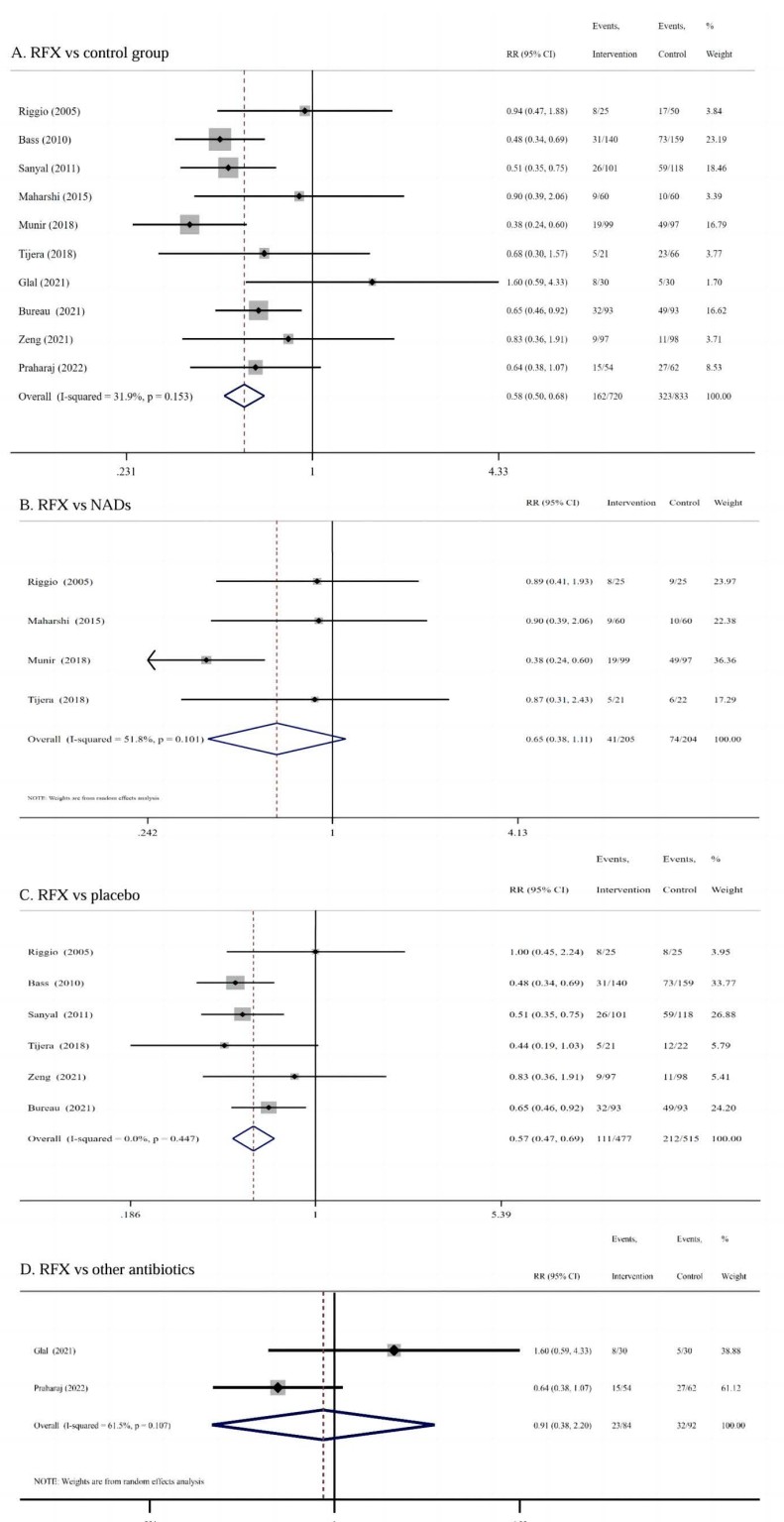

**Fig 2. Meta Analysis for HE Incidence.** Compared with the control group, RFX treatment significantly reduced the incidence of HE (RR = 0.58) **(A)**; Subgroup analysis indicated RFX has noncomparable effects to NADs in reducing the risk of HE occurrence (RR = 0.65) **(B)**, but more effective than placebo (RR = 0.57) **(C)**.

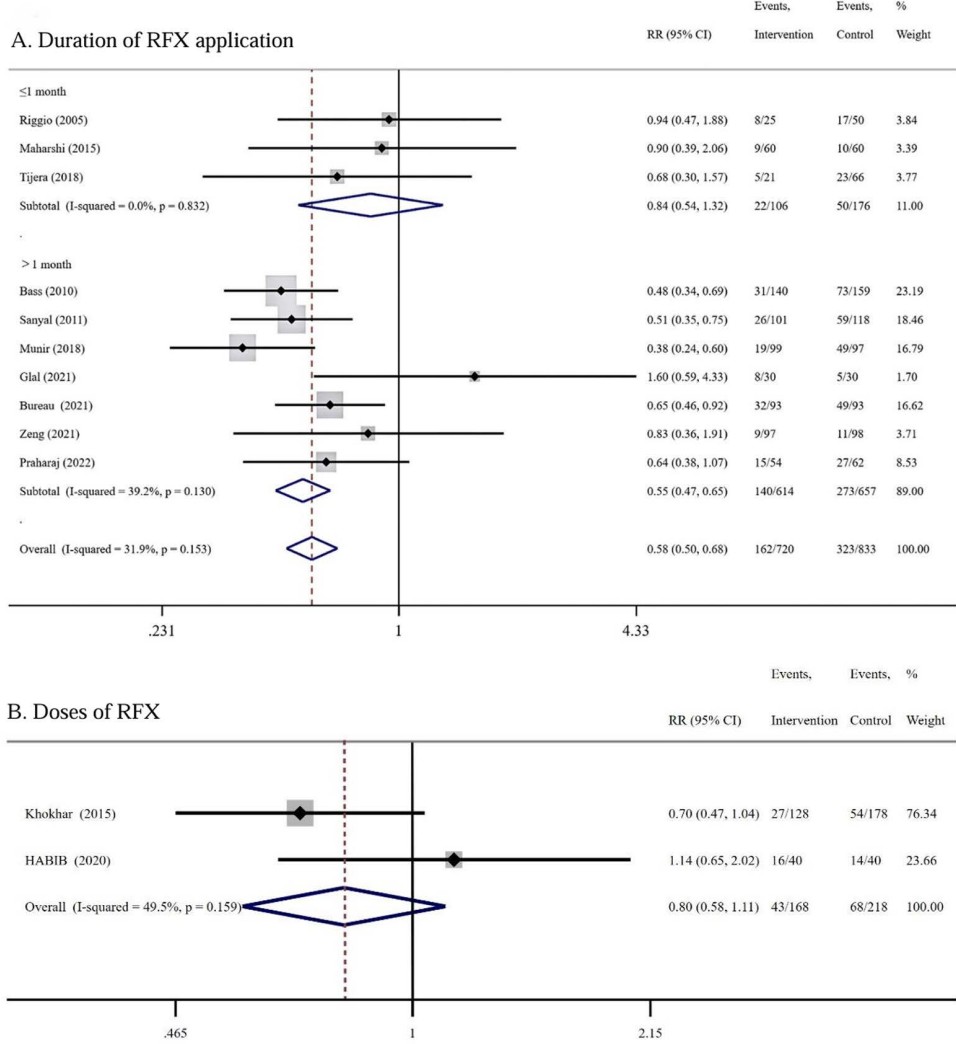

**Fig 3. Subgroup Meta Analysis for HE Incidence Based on Treatment Duration and Dose.** RFX treatment was more effective in the treatment duration >1 month subgroup (RR = 0.55), but no statistically significant in the treatment duration ≤1 month subgroup(A); The difference in HE incidence between the RFX low-dose group (≤550 mg/d) and conventional dose group (1100 mg/d) was not statistically significant (RR = 0.80) **(B)**.

**Secondary outcomes. Effect of RFX on blood ammonia:** 3 RCTs with a total of 287 patients reported on the effect of RFX on blood ammonia levels, with 133 patients in the RFX group and 154 patients in the group receiving NADs, nitazoxanide, and placebo respectively. Blood ammonia levels were higher in patients with cirrhosis after RFX treatment than in the controls (SMD = 0.25, 95% CI: 0.01–0.48, $I^2$ = 0.0%, $P$ = 0.041) (S3A Fig).

**Hospitalization rate:** 2 RCTs reported hospitalization rate due to underlying disease or HE episodes during RFX treatment. Meta-analysis demonstrated that RFX treatment was more beneficial in lowering the hospitalization rate in patients with cirrhosis compared with the control group (RR = 0.58, 95% CI: 0.37–0.88, $I^2$ = 0.0%, $P$ = 0.012) (S3B Fig).

**Publication bias.** Publication bias was performed on 10 RCTs reporting the effect of RFX versus non-RFX/placebo on the prevention of HE. The funnel plot was asymmetrical, suggesting possible publication bias (S4 Fig), and Egger's test confirmed the presence of publication bias ($P$ = 0.028). The pooled conclusion did not change after adjustment for

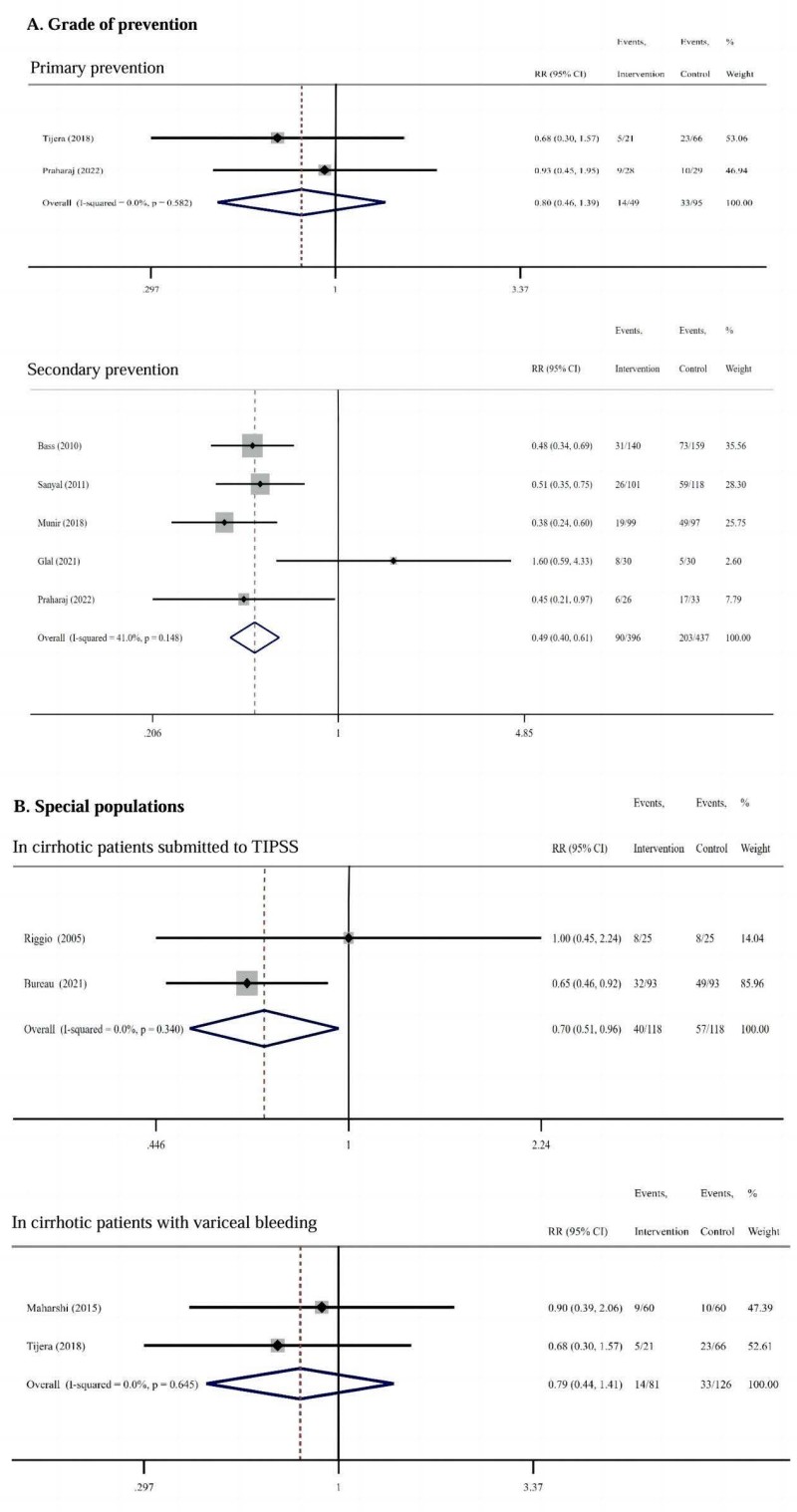

**Fig 4. Meta Analysis for Effectiveness of RFX for Primary and Secondary Prevention.** RFX was mild comparable to the controls in primary prevention of HE occurrence (RR = 0.80), but with a greater advantage over the controls in secondary prevention by reducing the risk of HE recurrence. (RR = 0.49) **(A)**; RFX had a beneficial effect on reducing the risk of HE after TIPSS compared with placebo (RR = 0.70), but only mild comparable in preventing HE in cirrhotic patients with variceal bleeding with the controls (RR = 0.79) **(B)**.

potential publication bias by using the trim and fill method (before: logRR = 0.535 95% CI: 0.693–0.376, *P* = 0.000, after: logRR = 0.535, 95% CI: 0.429–0.667, *P* = 0.000), suggesting the result was stable.

## Discussion

HE is one of the serious complications of cirrhosis which impairs consciousness, and even leads to death, resulting in a poor prognosis for patients. The presence of cirrhosis is associated with gut dysbiosis, alteration of intestinal barrier permeability, and release of endotoxin, which may trigger development of a profound inflammatory state and exacerbate the imbalance of ammonia metabolism. These changes are felt to be key to the development of HE, and the beneficial effect of RFX on the gut microbiome can interfere with these pathological processes [24,25]. The treatment and prevention of HE focuses on empirical treatment aimed at reducing blood levels of ammonia, and RFX can be an influential drug of choice by slowing ammonia creation and uptake in the gut. It has been demonstrated that RFX significantly reduce HE grade, improve cognitive impairment and long-term outcomes in patients with HE [26]. Nevertheless, there is ongoing debate concerning the application of RFX in the prevention of hepatic encephalopathy. For instance, the dose and duration of RFX application are uncertain, and the benefits and risks when compared to other medications are still unclear. Hence, higher-level evidence is required to support the clinical implementation of RFX for the prevention of HE. This Meta-analysis found evidence that RFX has advantage over the non-RFX/placebo in preventing of HE in patients with cirrhosis, and this effect increases with the prolongation of treatment duration.

In this Meta-analysis, we evaluated the effect of RFX compared with NADs, other antibacterial drugs, and placeboes in the prevention of HE in patients with cirrhosis. It was shown that RFX is noncomparable to NADs and other antibacterial drugs (nitazoxanide and norfloxacin), but is more advantageous than placebo in preventing HE. The key to primary prevention of HE in cirrhotic patients is to treat primary liver disease and deal with potential predisposing factors, and there is no effective drug for primary prevention. It is an updated Meta-analysis to evaluate the effect of RFX in the primary prevention of HE, and the result shows that the preventive effect of RFX on HE is not superior to that of the control group. The incidence of HE recurrence is 40% in 1 year in patients with cirrhosis [27], which seriously threatens the life safety of patients. However, this Meta-analysis supports that RFX is favorable for the secondary prevention of HE, which is consistent with the findings of two previous Meta-analysis [11,28]. Moreover, another highlight of this Meta-analysis is stratified analysis according to patient characteristics. RFX reduces the risk of developing HE after TIPSS compared with placebo, which is not consistent with the recent Meta-analysis [29]. The inconsistency between the two studies may be due to the more stringent study design and control group setting in our study. And we assessed the effectiveness of RFX for HE prophylaxis after variceal bleeding in cirrhosis, and the result showed that RFX was not superior to non-RFX and placebo. However, only Maharshi et al [17] and Tijera et al [18] were included, and the sample size was modest, thus more RCTs with larger samples are needed in the future.

Currently, the recommended dose of RFX for the prevention and treatment of HE in patients with cirrhosis is mostly 1100–1200mg/d [30,31]. Among the included 14 RCTs, 12 RCTs with non-RFX and placebo controls reported RFX daily doses also concentrated at 1100 mg and 1200 mg. However, the data were not further pooled for evaluation, because confounding factors such as the type of control drug, frequency of administration, duration of treatment, and the subject's underlying disease condition may interfere with results. Only Khokhar et al [16] and Habib et al [19] reported the difference in the effectiveness of different doses of RFX in preventing HE, and the Meta-analysis demonstrated that low doses of RFX may be as effective as conventional doses in preventing HE. Interestingly, a multi-center randomized open-labelled prospective study [22] investigated that 800mg/d RFX significantly decreased the occurrence of HE in patients with decompensated cirrhosis. Therefore, more evidence from large prospective studies is expected in the future.

Finally, the results of this Meta-analysis showed that there was no statistically significant difference in the incidence of adverse events such as dizziness or headache, abdominal pain, diarrhea, nausea or vomiting, bloating or abdominal discomfort, constipation, rash between RFX and the control group. However, it is worth noting that a subgroup analysis

showed a lower incidence of diarrhea with RFX applications compared with NADs. No RFX-related serious adverse events were reported in any of the original studies included in this Meta-analysis. It is suggested that RFX is a safe and effective drug option for the prophylaxis of HE. The long-term application of RFX to induce altered intestinal flora resistance has been a focused academic concern in recent years [32,33]. A prospective study demonstrated that prophylactic administration of RFX for 12 weeks in patients with cirrhosis improved hyperammonemia and neurophysiological function, maintained gut microbiota diversity, composition, and did not change the overall resistome [34]. But there is also literature that reports a high percentage of clostridium difficile infection (CDI) cases were detected in cirrhotic patients receiving RFX [35]. Unfortunately, the literature included in this Meta-analysis lacks descriptions and comparisons of the adverse effects of RFX-induced intestinal flora resistance. Only Bass et al [13] with two patients developed CDI during RFX treatment, but the two patients had high-risk factors for CDI such as advanced age, multiple antibiotics during hospitalization, and a history of proton pump inhibitor therapy. In particular, two of the patients did not discontinue RFX while receiving anti-infective treatment for CDI and then recovered, so it is difficult to conclude whether CDI was associated with RFX.

This systematic review on the efficacy and safety of RFX for the prevention of HE in patients with cirrhosis found sufficient evidence to reach conclusions and make recommendations for clinical practice. But it can not be ignored that this study has several limitations. Firstly, the unpublished literature was not searched for in this study, which may increase the risk of publication and reporting bias. Secondly, to improve the accuracy of the analysis, only RCTs were included in this Meta-analysis, six studies were not blinded to participants and personnel, and the risk of blinding in outcome assessment of 12 studies was unclear, which potentially affects the stability of the results. The RFX interventions were not entirely consistent across the included RCTs, such as drug dose and duration of treatment, which may have influenced the results. In addition, in this Meta-analysis, significant heterogeneity existed in outcomes including the comparison of RFX with other antibacterial drugs for HE prevention, the secondary prevention of HE, and the incidence of abdominal distension or abdominal discomfort. It is unfortunate that because of the limited information provided in the original study, further subgroup analysis and Meta-regression were not used to identify sources of heterogeneity.

In conclusion, identifying an optimal medicine to prevent HE is critical for improving the prognosis of patients with cirrhosis. It is crucial to conduct scientific and accurate assessments of the efficacy and safety of drugs in order to guide clinical application. This systematic review focuses on summarizing and analyzing the effectiveness of RFX for the prevention of HE in patients with cirrhosis to help clinicians in their medication decisions. The results suggested that the risk of HE occurrence in patients with cirrhosis decreases with a longer duration of RFX treatment.

## Key points

- Evidence on RFX application for preventing HE in patients with cirrhosis is limited and higher-grade evidences are required.

- RFX had a beneficial effect on the prevention of HE (RR=0.58, 95% CI: 0.50-0.68), with noncomparable effects to NADs (RR=0.65, 95% CI: 0.38-1.11), and more effective than placebo (RR=0.57, 95% CI: 0.47-0.69). And RFX helped to reduce the incidence of HE after TIPSS (RR=0.70, 95% CI: 0.51-0.96).

- Large-scale, randomized, prospective cohort studies are needed to provide reliable data on the effect of RFX dose on the prevention of HE in cirrhosis.

## Supporting Information

**S1 File. PRISMA check list.**
(DOCX)

**S2 File. Table includes all literature and exclusion grounds.**
(XLSX)

**S3 File. Supplementary Figure S1-4.** Fig S1 Cochrane Risk of Bias Assessment. Risk of bias graph (A); Risk of bias summary (B). 12 RCTs described the random sequence generation and manifested a low risk of attrition, reporting or other biases, 6 RCTs described the allocation concealment, and 5 RCTs described the blinding of participants and personnel as low risk. Fig S2 The forest plot of the effect of RFX treatment on all-cause mortality. No statistical difference in mortality risk between both overall and subgroup analyses. Fig S3 The forest plot of the effect of rifaximin treatment on blood ammonia and hospitalization rate. Blood ammonia levels after RFX treatment were mild higher than in the control group (including NADs, nitazoxanide, and placebo) (A), while more beneficial in lowering the hospitalization rate compared with the control group (B). Fig S4 Funnel diagram of RFX for prevention of HE. Funnel plot was asymmetrical that suggesting the possible publication bias.
(DOCX)

**S4 File. Supplementary Table S1-2.** Table S1 The basic information of selected literature. Table S2 The incidence of adverse events during RFX treatment.
(DOCX)

## Author contributions

**Conceptualization:** Yadong Wang.

**Data curation:** Yangyang Hu, Xing Zhang, Ying Xiao, Zhinian Wu.

**Methodology:** Yangyang Hu.

**Writing – original draft:** Yangyang Hu, Xing Zhang.

**Writing – review & editing:** Yadong Wang.

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
