## [Decision Letter · Decision Letter 0]

29 Oct 2024

PONE-D-24-31116Efficacy and safety of Rifaximin in preventing hepatic encephalopathy: a Systematic Review and Meta-AnalysisPLOS ONE

Dear Dr. Wang,

Thank you for submitting your manuscript to PLOS ONE. After careful consideration, we feel that it has merit but does not fully meet PLOS ONE’s publication criteria as it currently stands. Therefore, we invite you to submit a revised version of the manuscript that addresses the points raised during the review process.

We look forward to receiving your revised manuscript.

Kind regards,

Peter Starkel, M.D., Ph.D.

Academic Editor

PLOS ONE

Journal requirements: When submitting your revision, we need you to address these additional requirements. 1. Please ensure that your manuscript meets PLOS ONE's style requirements, including those for file naming. The PLOS ONE style templates can be found at https://journals.plos.org/plosone/s/file?id=wjVg/PLOSOne_formatting_sample_main_body.pdf and https://journals.plos.org/plosone/s/file?id=ba62/PLOSOne_formatting_sample_title_authors_affiliations.pdf 2. Thank you for stating the following financial disclosure:  [This study was supported by grants from the Natural Science Foundation of Hebei Province of China (H2023206042).].  Please state what role the funders took in the study.  If the funders had no role, please state: ""The funders had no role in study design, data collection and analysis, decision to publish, or preparation of the manuscript."" If this statement is not correct you must amend it as needed. Please include this amended Role of Funder statement in your cover letter; we will change the online submission form on your behalf. 3.We note that your Data Availability Statement is currently as follows: [All relevant data are within the manuscript and its Supporting Information files.] Please confirm at this time whether or not your submission contains all raw data required to replicate the results of your study. Authors must share the “minimal data set” for their submission. PLOS defines the minimal data set to consist of the data required to replicate all study findings reported in the article, as well as related metadata and methods (https://journals.plos.org/plosone/s/data-availability#loc-minimal-data-set-definition). For example, authors should submit the following data: - The values behind the means, standard deviations and other measures reported;- The values used to build graphs;- The points extracted from images for analysis. Authors do not need to submit their entire data set if only a portion of the data was used in the reported study. If your submission does not contain these data, please either upload them as Supporting Information files or deposit them to a stable, public repository and provide us with the relevant URLs, DOIs, or accession numbers. For a list of recommended repositories, please see https://journals.plos.org/plosone/s/recommended-repositories. If there are ethical or legal restrictions on sharing a de-identified data set, please explain them in detail (e.g., data contain potentially sensitive information, data are owned by a third-party organization, etc.) and who has imposed them (e.g., an ethics committee). Please also provide contact information for a data access committee, ethics committee, or other institutional body to which data requests may be sent. If data are owned by a third party, please indicate how others may request data access. 4. Please include captions for your Supporting Information files at the end of your manuscript, and update any in-text citations to match accordingly. Please see our Supporting Information guidelines for more information: http://journals.plos.org/plosone/s/supporting-information.  5. We notice that your supplementary figures are uploaded with the file type 'Other'. Please amend the file type to 'Supporting Information'. Please ensure that each Supporting Information file has a legend listed in the manuscript after the references list. 6. As required by our policy on Data Availability, please ensure your manuscript or supplementary information includes the following:  A numbered table of all studies identified in the literature search, including those that were excluded from the analyses.   For every excluded study, the table should list the reason(s) for exclusion.   If any of the included studies are unpublished, include a link (URL) to the primary source or detailed information about how the content can be accessed.  A table of all data extracted from the primary research sources for the systematic review and/or meta-analysis. The table must include the following information for each study:  Name of data extractors and date of data extraction  Confirmation that the study was eligible to be included in the review.   All data extracted from each study for the reported systematic review and/or meta-analysis that would be needed to replicate your analyses.  If data or supporting information were obtained from another source (e.g. correspondence with the author of the original research article), please provide the source of data and dates on which the data/information were obtained by your research group.  If applicable for your analysis, a table showing the completed risk of bias and quality/certainty assessments for each study or outcome.  Please ensure this is provided for each domain or parameter assessed. For example, if you used the Cochrane risk-of-bias tool for randomized trials, provide answers to each of the signalling questions for each study. If you used GRADE to assess certainty of evidence, provide judgements about each of the quality of evidence factor. This should be provided for each outcome.   An explanation of how missing data were handled.   This information can be included in the main text, supplementary information, or relevant data repository. Please note that providing these underlying data is a requirement for publication in this journal, and if these data are not provided your manuscript might be rejected.  

Reviewers' comments:

Reviewer's Responses to Questions

**Comments to the Author**

1. Is the manuscript technically sound, and do the data support the conclusions?

Reviewer #1: Partly

2. Has the statistical analysis been performed appropriately and rigorously? 

Reviewer #1: Yes

3. Have the authors made all data underlying the findings in their manuscript fully available?

Reviewer #1: Yes

4. Is the manuscript presented in an intelligible fashion and written in standard English?

Reviewer #1: Yes

5. Review Comments to the Author

Reviewer #1: In this paper, Hu and colleagues conducted a meta-analysis of the efficacy and safety of rifaximin in the prevention of hepatic encephalopathy (HE).

The main problem with this paper is that the authors do not clearly define the concept of HE prevention. Is it primary or secondary prevention? This lack of definition makes the manuscript confusing and not easy to read from the abstract to the conclusion.

Other important comments:

-I do not understand why the authors used "fulminant hepatic failure with cerebral oedema" as a keyword?

-Study selection section: the authors mentioned that the patients had CLD "mainly consisting of cirrhosis". What are the other forms of chronic liver disease? I think it should be only cirrhotic patients. The authors should clarify this.

-I do not have access to Figure S1.

-The authors mentioned LOLA in the introduction and study design, but there are no data on LOLA in the results. Authors should clarify.

-There is no legend for the figure.

Minor comments:

-In the abstract: NADs is not defined.

-Authors should use TIPSS and not TIPS.

-In the secondary outcomes section, authors should modify “3.4.2.2 Hospitalization rate”.

6. PLOS authors have the option to publish the peer review history of their article (what does this mean? ). If published, this will include your full peer review and any attached files.

**Do you want your identity to be public for this peer review?** For information about this choice, including consent withdrawal, please see our Privacy Policy .

Reviewer #1: No

---

## [Author Response · Author response to Decision Letter 1]

6 Feb 2025

Dear Editor,

Thank you very much for your decision letter and advice on our manuscript (PONE-D-24-31116) entitled “Efficacy and safety of Rifaximin in preventing hepatic encephalopathy: a Systematic Review and Meta-Analysis”. We also thank the reviewers for the constructive comments and suggestions. We have revised the manuscript accordingly, and all amendments are indicated by red font in the revised manuscript. In addition, our point-by-point responses to the comments are listed below this letter.

We hope that our revised manuscript is now acceptable for publication in your journal and look forward to hearing from you soon.

With best wishes,

Yours sincerely,

Yadong Wang et al.

First of all, we would like to express our sincere gratitude to the reviewers for their constructive and positive comments. In the past two months, we have updated our search methods based on the suggestions and doubts of reviewers to ensure the correctness of our study. The results and descriptions have been supplemented or revised in the revised manuscript. The following is our point-to-point replies to the questions raised by reviewers, please review.

Replies to Reviewer 1

Reviewer #1: In this paper, Hu and colleagues conducted a meta-analysis of the efficacy and safety of rifaximin in the prevention of hepatic encephalopathy (HE).

The main problem with this paper is that the authors do not clearly define the concept of HE prevention. Is it primary or secondary prevention? This lack of definition makes the manuscript confusing and not easy to read from the abstract to the conclusion.

Reply: Thank you for reviewing this article and constructive comments. I 'm sorry to misinterpret your presentation. Our meta-analysis included all articles, including primary and secondary prevention, and to avoid misunderstandings, we added the corresponding location in the revised manuscript for your review. The study showed that RFX was not superior to the control group in the primary prevention of HE. However, this meta-analysis more supports that RFX is beneficial for the secondary prevention of HE, especially in stratified analysis confirming that RFX reduces the risk of HE after TIPS. These results are also described in the ‘Discussion’ section. Given your suggestion to increase article readability and avoid logical confusion, we readjust some of the wording in the original text based on your comments. The corresponding contents are marked in ‘Revised Manuscript with Track Changes’ and please review them again.

Other important comments:

-I do not understand why the authors used "fulminant hepatic failure with cerebral oedema" as a keyword?

-Study selection section: the authors mentioned that the patients had CLD "mainly consisting of cirrhosis". What are the other forms of chronic liver disease? I think it should be only cirrhotic patients. The authors should clarify this.

-I do not have access to Figure S1.

-The authors mentioned LOLA in the introduction and study design, but there are no data on LOLA in the results. Authors should clarify.

-There is no legend for the figure.

Reply: Thank you very much for your professional comments on our article. In view of your five questions above mentioned, the explanations are as follows:

-We apologize that our negligence increases your misunderstanding, so we intend to include all HE patients initially in this study, including acute liver failure (previously called fulminant liver failure), which is also to avoid that incomplete literature search may affect the final results of meta-analysis. It was confirmed that no fulminant hepatic failure-related studies were included in the meta-analysis literature, so it was deleted in the revised manuscript. Please review here.

-We sincerely thank you for your valuable feedback to improve the quality of our manuscript. As you say, HE is a syndrome of psycho-behavioral abnormalities in patients with liver failure or cirrhosis caused by different triggers. It was verified again that there were indeed two articles in the studies included in this article that the authors described CLD, but the authors did not specify the category of so-called CLD. Therefore, according to your professional suggestions, we eliminated those two literatures and re-performed Meta-analysis, so that all the subjects were patients with liver cirrhosis. We also revised the results of the latest meta-analysis of the included literature and reviewed multiple corresponding contents in the text (e.g. 'Study selection and Introduction data extraction' and so on) were modified to replace CLD with cirrhosis, please review again.

-We are very sorry for not being able to access Figure S1.We scrutinized the content and order of all pictures uploaded and corrected these errors in a timely manner. Thank you again for careful review.

-Because LOLA is a common medicine to prevent/treat HE in patients with liver disease. Therefore, during the literature search, randomized controlled studies using other interventions (e.g. NADs, other antibiotics, LOLA, and placebo) were included in order to compare the differences between RFX and these agents in the prevention of HE. This is also in line with the study objectives of this paper. However, no literature was found comparing RFX with LOLA by appropriately and rigorously 's search strategy. We also respect the search results. The results of subgroup analyses for the remaining interventions (e.g. NADs, other antibiotics, and placebo) are shown in Figure 2. Please review again.

-We apologize for the confusion caused by figures. Figures’ notes have been added at corresponding locations. Please review again.

Minor comments:

-In the abstract: NADs is not defined.

-Authors should use TIPSS and not TIPS.

-In the secondary outcomes section, authors should modify “3.4.2.2 Hospitalization rate”.

Reply: Thanks to your meticulous review. We apologize for the inconvenience caused by our negligence.

-NADs means nonabsorbent disaccharides, mainly including lactulose and lactitol. Following your suggestion, we illustrate the category of NADs within ‘()’ after the first 'NADs'. This is more conducive to readers' reading and understanding. Thank you for your advice, please review again.

-We searched the literature and found that there were two expressions, transjugular intrahepatic portosystemic shunt (TIPS) and transjugular intrahepatic portosystemic stent shunt (TIPSS), and TIPSS was more accurate, and we have corrected all "TIPS" in the text to "TIPSS". Please review again.

-Thank you for your meticulous and patient review and apologize for my negligence, and we have revised the original text and painstakingly reviewed it again

---

## [Editor Report · Decision Letter 1]

8 Apr 2025

Efficacy and safety of Rifaximin in preventing hepatic encephalopathy: a Systematic Review and Meta-Analysis

PONE-D-24-31116R1

Dear Dr. Wang,

We’re pleased to inform you that your manuscript has been judged scientifically suitable for publication and will be formally accepted for publication once it meets all outstanding technical requirements.

Kind regards,

Peter Starkel, M.D., Ph.D.

Academic Editor

PLOS ONE

Additional Editor Comments (optional):

Reviewers' comments:

The authors have addressed my comments. The paper has been improved and the changes and corrections made to the paper make it more sound with a message that is closer to the clinical reality. For me it is suitable for publication.

---

## [Editor Report · Acceptance letter]

PONE-D-24-31116R1

PLOS ONE

Dear Dr. Wang,

I'm pleased to inform you that your manuscript has been deemed suitable for publication in PLOS ONE. Congratulations! Your manuscript is now being handed over to our production team.

Kind regards,

on behalf of

Dr Peter Starkel

Academic Editor

PLOS ONE